# Vancomycin-Resistant Enterococci (VRE) in Nigeria: The First Systematic Review and Meta-Analysis

**DOI:** 10.3390/antibiotics9090565

**Published:** 2020-09-01

**Authors:** Yusuf Wada, Azian Binti Harun, Chan Yean Yean, Abdul Rahman Zaidah

**Affiliations:** 1Department of Medical Microbiology and Parasitology, School of Medical Sciences, Universiti Sains Malaysia, Kubang Kerian 16150, Malaysia; wadayusuf34@gmail.com (Y.W.); azian@usm.my (A.B.H.); yeancyn@yahoo.com (C.Y.Y.); 2Department of Zoology, Faculty of Life Sciences, Ahmadu Bello University, Zaria 810211, Nigeria; 3Hospital Universiti Sains Malaysia, Universiti Sains Malaysia, Kubang Kerian 16150, Malaysia

**Keywords:** *Enterococcus*, vancomycin resistance, systematic review, meta-analysis, Nigeria

## Abstract

Vancomycin-Resistant Enterococci (VRE) are on the rise worldwide. Here, we report the first prevalence of VRE in Nigeria using systematic review and meta-analysis. International databases MedLib, PubMed, International Scientific Indexing (ISI), Web of Science, Scopus, Google Scholar, and African journals online (AJOL) were searched. Information was extracted by two independent reviewers, and results were reviewed by the third. Two reviewers independently assessed the study quality using the Preferred Reporting Items for Systematic Reviews and Meta-Analysis (PRISMA) checklist. OpenMeta analyst was used. The random effect was used, and publication bias was assessed using a funnel plot. Between-study heterogeneity was assessed, and the sources were analysed using the leave-one-out meta-analysis, subgroup analysis, and meta-regression. Nineteen studies met the eligibility criteria and were added to the final meta-analysis, and the study period was from 2009–2018. Of the 2552 isolates tested, 349 were VRE, and *E. faecalis* was reported the most. The pooled prevalence of VRE in Nigeria was estimated at 25.3% (95% CI; 19.8–30.8%; *I*^2^ = 96.26%; *p* < 0.001). Between-study variability was high (*t*^2^ = 0.011; heterogeneity *I*^2^ = 96.26% with heterogeneity chi-square (Q) = 480.667, degrees of freedom (df) = 18, and *p* = 0.001). The funnel plot showed no publication bias, and the leave-one-out forest plot did not affect the pooled prevalence. The South-East region had a moderate heterogeneity though not significant (*I*^2^ = 51.15%, *p* = 0.129). Meta-regression showed that all the variables listed contributed to the heterogeneity except for the animal isolate source (*p* = 0.188) and studies that were done in 2013 (*p* = 0.219). Adherence to proper and accurate antimicrobial usage, comprehensive testing, and continuous surveillance of VRE are required.

## 1. Introduction

*Enterococcus* is a Gram-positive and catalase-negative bacterium. It is an important gastrointestinal tract normal flora of most warm-blooded animals and humans [1,2]. However, different species of Gram-positive cocci could be an opportunistic pathogen causing various infectious diseases [3,4]. *Enterococcus* species especially *Enterococcus faecium* and *Enterococcus faecalis* are two common causes of urinary tract infection [5,6], inflammation of the lining of the heart and its valves, intra-abdominal abscesses, wound infections, bacteremia, and sepsis in human [7]. It has been proven that *Enterococcus* is the second leading cause of urinary tract and wound infections and the third leading cause of bacteremia in hospitals [8]. The inherent resistance to several antibiotics and their ability to cause infections has placed enterococci on the pedestal as an important hospital-acquired pathogen [9].

Hospital-acquired infection, especially that caused by Vancomycin-Resistant Enterococci (VRE), has been on the rise regardless of their low pathogenicity and virulence. VRE prevalence in the intensive care unit (ICU) of many hospitals worldwide is high and more so when patients have an underlying health condition such as diabetes mellitus, neutropenia, and impaired renal function [10]. In the treatment of infections caused by *Enterococcus*, vancomycin and sometimes with any other aminoglycoside, is used because of its bactericidal efficacy. These antibiotics are usually used to treat infections caused by methicillin-resistant *Staphylococcus aureus* and other Gram-positive bacteria [11,12]. Vancomycin is used as the last option in the treatment of *E**nterococcus* [9] as its resistance to antibiotics is as a result of either an inherent or acquired machinery. Isolates of *E. faecalis* and *E. faecium* exhibit high resistance to vancomycin while the reverse is the case for *E. gallinarum* and *E. flavescens* as they exhibit low resistance [13]. Genetic elements known as *van* genes confer resistance to *Enterococcus* of which *vanA* and *vanB* present mostly in *E. faecium* occur the most and are well-distributed, especially among hospital isolates [13]. There is a disturbing trend following several reports on the resistance of *enterococcus* to linezolid and daptomycin, two potent antibiotics used against VRE infection [14,15] while Melese et al. [16] stated that the persistent increase in nosocomial infection caused by VRE is being reported by several studies.

One of the most important goals of meta-analyses is to provide an accurate and reliable result by increasing the sample size and reducing the width of the 95% CI from the range of the various applicable studies. Several studies are reporting VRE in Nigeria as a result of its role in the livestock industry and the health sector. Nigeria is beginning to generate a lot of revenue from the livestock industry recently as a result of the border closure. This simply means that a lot of farmers would want to sell their product in time and might result in the use of growth promoters such as avoparcin. It is therefore important that this sector is closely guarded given the risk of importation of an infected and tainted product. The knowledge of VRE distribution can be used to develop a policy to curtail the spread of resistant bacteria while addressing the prevention, control, and treatment as it is of public health significance. Such a policy would ensure that healthy livestock products are consumed, and resistant bacteria monitored. It is, however, necessary to obtain the pool prevalence of VRE in Nigeria from different sources using meta-analysis to enable the Nigeria Center for Disease Control (NCDC) to develop a policy and road map for its prevention and elimination. A meta-analysis would help us validate the results of various studies reporting VRE in Nigeria and put forward a measure that is accurate and reliable.

It is based on the above points that this paper was designed to determine the pooled prevalence of VRE using a systematic literature review and meta-analysis in Nigeria.

## 2. Results

### 2.1. Search Results and Eligible Studies

Figure 1 shows the search results. A total of 500 studies were found, of which 120 were left after duplicates were removed. Of the 120 studies screened for eligibility, 97 were excluded as they did not meet any of the inclusion criteria. Twenty-three full-text articles were assessed for eligibility with four excluded since vancomycin was not used in their antimicrobial susceptibility test and had insufficient information. A total of 19 full-text studies were used for quantitative analyses.

### 2.2. Characteristics of the Eligible Studies

All the 19 studies included in this review were cross sectional by design. Most of the studies were reported from the South-West region (*n* = 8) [17,18,19,20,21,22,23,24]. Other studies include the North-Central region (*n* = 3) [25,26,27], South-East region (*n* = 3) [28,29,30], South-South (*n* = 4) [31,32,33,34], and North-West region (*n* = 1) [35]. No study was reported in the North-East region of the country. Of the 2552 isolates tested, 349 were VRE. The sample size ranges from as low as 7 [17] to as high as 658 [20] and prevalence as high as 88.9% in the South-South region [31] to as low as 1.1% in the North-Central region [27] (Table 1). The highest number of VRE (*n* = 77) was isolated from environmental sources in the study conducted in the South-West region of Nigeria. The study analysed the highest number of specimens compared to others [20]. Most of the studies utilised the disk diffusion method in their antimicrobial susceptibility testing except for [18] and [27], who utilised agar dilution and VRE chromogenic agar, respectively (Table 1). The majority of the data included in analyses were from clinical studies (*n* = 8) which involved clinical specimens, and environmental studies (*n* = 7) with others from animal studies (*n* = 4) (Table 1). Details of the characteristics of the included studies are summarized in (Table 1) below and a map showing the spatial distribution and number of studies of VRE in Nigeria is shown in Figure 2.

Only 12 studies reported the reported the prevalence of VRE according to species (Table 2). *E. faecalis* was the most reported with a prevalence of 62.98% (148/235) followed by *E. faecium* with a prevalence of 21. 70% (51/235)

### 2.3. The Pooled Prevalence of VRE

The pooled prevalence of VRE in Nigeria was estimated at 25.3% (95% CI; 19.8–30.8%; *I*^2^ = 96.26%; *p* < 0.001) (Figure 3). Random-effects meta-analyses were carried out using the total sample size and number of positives (effect size, standard error of effect size) to estimate the prevalence of VRE in Nigeria. Between-study variability was high (*t*^2^ = 0.011; heterogeneity *I*^2^ = 96.26% with heterogeneity chi-square (*Q*) = 480.667, degrees of freedom (df) = 18, and *p* = 0.001).

Sensitivity analysis using the leave-one-out forest plot revealed that no single study significantly influenced the heterogeneity and pooled prevalence of VRE (25.3%; 95% CI; 19.8–30.8%; *p* < 0.001) (Figure 4). The presence of publication bias was observed from the drawn asymmetric funnel plot (Figure 5) which indicates no publication bias.

### 2.4. Subgroup Meta-Analysis

Since this meta-analysis showed substantial heterogeneity, subgroup analysis was done using the study period, study area, isolate sources, and detection method to identify the possible sources of heterogeneity among the studies. The result of subgroup meta-analysis by study region revealed overall large variability in studies reporting the prevalence of VRE (the Higgins *I*^2^ statistic = 96.26% with heterogeneity chi-square (*Q*) = 480.667, degrees of freedom = 18, and *p* < 0.001). The Southeast region was the only region with moderate heterogeneity, though not significant (*I*^2^ = 51.15%, *p* = 0.129), revealing a probable cause of heterogeneity. The overall statistics are shown in Table 3.

Similarly, the result of subgroup meta-analysis by isolate source revealed the highest variability in isolates from clinical (*I*^2^ = 195.18%), followed by the environment (*I*^2^ = 95.33%), and animal sources (*I*^2^ = 96.37%). Isolates from environmental sources had the highest prevalence (27.2%, CI 17.3–13.2%). The overall statistics are shown in Table 4.

Further, the result of subgroup meta-analysis by the detection method revealed that 16 of the studies utilised the disc diffusion method in their antimicrobial susceptibility test accounting for a prevalence of 33.8% with a CI of 24.3–43.4% and *I*^2^ of 93.84%. The overall statistics are shown in Table 5.

Finally, the result of subgroup meta-analysis by study period revealed that the years 2017 and 2018 had four studies each, and the study period ranged from 2009 to 2018. The prevalence of VRE ranged from 42.9%, CI 6.2–79.5% in 2009 to 53.6%, CI 26.5–80.7%, indicating an increase in the prevalence of VRE over 10 years. The overall statistics are shown in Table 6.

### 2.5. Meta-Regression

Meta-regression analysis was done for each variable included in the study individually. The variables included were study region, study year, isolate sources, and detection method. Continuous variables were subjected to assessment to observe a linear relationship with the independent effect size. Variables with *p*-values < 0.25 were used in the multivariable meta-regression analysis. Independent variables such as study region, study year, isolate sources, and detection method had a reasonably significant value and were retained in the final multivariate analysis. Most of the variables were significantly associated with the prevalence of VRE in the final multivariate meta-regression except for the animal isolate source (*p* = 0.188) and studies done in 2013 (*p* = 0.219). Interestingly, all the variables listed contributed to the heterogeneity observed in this study except for the animal isolate source and studies done in 2013. No result was computed for the study period 2016–2018. The final multivariate meta-regression is shown in Table 7 below.

## 3. Discussion

To the best of our knowledge, this is the first study to determine the prevalence of VRE in Nigeria using systematic review and meta-analysis. For interventions to be accurately formulated, the prevalence of VRE needs to be known. The NCDC has a program in collaboration with the Federal Ministries of Health, Agriculture and Rural Development and Environment designed to checkmate antimicrobial resistance (AMR) [36]. A National AMR Technical Working Group (AMR-TWG) was inaugurated with members from several sectors such as human and animal health, food animal production, and the environment. The objective of this group is to analyse the situation of AMR in Nigeria and to develop an action plan for its prevention treatment and control. This is a robust plan, but the plan did not list VRE as a priority pathogen given that VRE is one of the most common causes of nosocomial infection worldwide.

The results presented in this report were from the analysis of data obtained through a systematic review of scientific publications on the prevalence of VRE at the country level between the years of 2009–2019, and the literature was heterogeneous. This review did not only take into consideration VRE in clinical settings but also in animals and the environment to get a holistic picture of VRE in Nigeria. The final meta-analysis of the prevalence was done only on 19 articles.

The random effect meta-analysis result showed high variability with Higgin’s *I*^2^, which indicates that the variability between studies was not as a result of chance alone. The detection method, study region, study period, and clinical and environmental isolate source were highly significant predictors of the prevalence of VRE, indicating that these variables explain a substantial portion of the variability between studies. However, the animal isolate source and studies done in 2013 retained in the final meta-regression seem statistically insignificant in explaining the study variability.

Although considerable methodological differences between studies existed, these differences were pooled for this review. Therefore, the pooled prevalence of VRE in Nigeria was estimated at 25.3%. This indirectly indicates the potential existence of VRE, not only in health care settings but in in the environment as well as animals in Nigeria, and its likely spread to communities unless properly contained. Our estimate is comparable with reports from Malaysia 25% [37] but higher than those reported in Ethiopia 14.8% [16], Iran 14%, 18.75% [38,39], North America (21%), Asia (24%), Europe (20%) [40], Germany (9.8%) [41], Iran (9.4%) [42], the United Kingdom (9.2%) [43], and Singapore (9.3%) [44]. Our estimate was probably higher because our studies included animal and environmental sources in addition to clinical settings unlike all the studies listed above where they largely centred on clinical settings. This high prevalence could be as a result of various risk factors such as contact with VRE patients, infected animals, surfaces and objects, underlying conditions, serious illness, prior hospitalization, use of catheters, and improper antibiotic usage [45]. Camins et al. [46] stated that health care contacts were the likely source of VRE colonization and infection, and this is plausible in situations where infection control knowledge, attitudes, and practices among healthcare workers, farmworkers, and the general population are poor in third-world countries [47,48,49,50] and Nigeria [26]. The antimicrobial susceptibility testing mainly relied on the disc diffusion method and was interpreted according to the Clinical and Laboratory Standard Institute (CLSI) guideline. However, agar dilution and VRE chromogenic agar were also used. Another study in Iran by Shokoohizadeh et al. [51] reported a 48.9% higher prevalence in patients hospitalised than this study estimate. Adams et al. [52] stated that the prevalence of VRE tends to be higher in critically ill and hospitalised patients than n non-hospitalised patients, unlike this present study where isolate sources were diverse. Another probable reason might be the study period as these studies were mostly done in the 1990s and 2000’s following the first reports of VRE [53,54], while the oldest study from our analysis was in 2009 and the earliest in 2019 where a ban was already placed on the indiscriminate use of vancomycin [55].

Results obtained from this review indicate that *E. faecalis* is the most reported VRE and this might be because most enterococcal infections are caused by *E. faecalis* [56] and can be treated with aminoglycosides and beta lactams. Because resistance to vancomycin is more regular in homogenous *E. faecalis*, administering of beta lactams should be at the forefront before the use of conventional culture [56]. Arias and Murray [57] and Davis et al. [58] stated that most VRE infections were caused by *E. faecalis*. This has changed as more VRE caused by *E. faecium* are increasingly being reported [59,60,61] because of their resistance to different group of antibiotics which are mostly expensive [62,63]. In *E. faecalis*, however, there is a marked difference in the occurrence and nature of resistance [64,65] even though *E faecalis* exhibit some level of acquired resistance.

Prevalence of VRE based on study area or region was also estimated. The highest estimated prevalence was in the South-South (56.2%) which is more than twice that estimated in the South-West (20.7%), North-Central (22.4%), South-East (10.2%) and North-West (25.0%). These regional differences could be ascribed to the type of environment and animal samples obtained, study period, the disparity in antibiotic use, detection method, and specimen type. No study or estimate was reported in the North-East of Nigeria. This is probably because this region has been plagued by insecurity as a result of insurgency which could deter researchers from conducting research [66].

According to data analyses for the isolate source, VRE prevalence was high in clinical, animal, and environmental sources and this is worrisome. These estimates indicate the depth of the spread of VRE in Nigeria and is one of the strengths of this study. Conducting these studies in different regions and isolates sources has provided a subtle picture of the prevalence in Nigeria. This does not give an in-depth explanation of the status of VRE in Nigeria, but it can be used as baseline information in its control.

In addition to obtaining isolates from different sources as strength of our study, our study also included a rigorous search with precise inclusion and exclusion criteria and we also observed frequently used specimens and methods of susceptibility testing. Several limitations also existed and these were our inability to report pooled estimates of VRE at the species level due to the limited number of included studies reporting enterococci at the species level, unavailability of studies from the Northeast region of Nigeria, which throws more question on the exact status of VRE in Nigeria, non-use of unpublished reports, and finally, the protocol of our study was not registered in PROSPERO.

## 4. Materials and Methods

### 4.1. Study Design and Protocol

The protocol of this study was designed according to the Preferred Reporting Items for Systematic Reviews and Meta-Analysis Protocol (PRISMA-P 2015) guidelines [67] (Appendix A). The risk of bias across studies and the risk of bias graph are presented in Appendix A.

### 4.2. Literature Review

A systematic review and meta-analysis were performed first by searching the PROSPERO database and database of abstracts of reviews of effects (DARE) (http://www.library.UCSF.edu) to check whether published or ongoing projects exist related to the topic. The literature search strategy, selection of studies, data extraction, and result reporting were done in accordance with the Preferred Reporting Items for Systematic Reviews and Meta-Analyses (PRISMA) guidelines. International databases MedLib, PubMed, ISI, Web of Science, Scopus, Google Scholar, and African journals online (AJOL) for published studies about the prevalence of VRE were also searched. PubMed was searched using the search strategy (“enterococcus”[MeSH Terms] OR “enterococcus”[All Fields]) AND (“nigeria”[MeSH Terms] OR “nigeria”[All Fields]), VRE [All Fields] AND (“nigeria”[MeSH Terms] OR “nigeria”[All Fields], (“epidemiology”[Subheading] OR “epidemiology”[All Fields] OR “prevalence”[All Fields] OR “prevalence”[MeSH Terms]) AND (“vancomycin-resistant enterococci”[MeSH Terms] OR (“vancomycin-resistant”[All Fields] AND “enterococci”[All Fields]) OR “vancomycin-resistant enterococci”[All Fields] OR (“vancomycin”[All Fields] AND “resistant”[All Fields] AND “enterococci”[All Fields]) OR “vancomycin resistant enterococci”[All Fields]) AND (“nigeria”[MeSH Terms] OR “nigeria”[All Fields]). Another search was also performed using keywords and their English equivalent (clinical infections, environmental VRE, VRE in poultry and farm animals, Gram-positive bacteria, enterococci, antibiotic resistance, glycopeptide, vancomycin, and Nigeria) with all possible combinations. Also, the titles and references from selected articles were an additional search tool. To reduce bias, the search process was conducted independently by two authors.

### 4.3. Inclusion and Exclusion Criteria for Studies

We considered all cross-sectional or cohort studies that reported the prevalence of vancomycin resistance in *Enterococcus* isolates or numbers of VRE and total enterococci isolates in patients suspected of having clinical infection, in poultry, poultry/animal product, farmworkers, and the environment in Nigeria. We also included studies in which the standard method was used to detect VRE and were published or reported in English.

Exclusion criteria for the analysis were as follows: studies with insufficient information; studies on antimicrobial susceptibility tests other than vancomycin (studies that did not include VRE), studies having fewer than two isolates, studies not reporting enterococcal isolates separately (no population denominator), reviews, comments and duplications, case report studies, and studies that did not report the prevalence of VRE.

### 4.4. Data Extraction

After studies were identified based on their eligibility criteria, the first author’s name, the publication year, the date of the study, the study location, the number of cases involved in the studies, the study method, the source of isolates, the sample size, and the prevalence of VRE infections were extracted from the manuscripts. Two independent reviewers extracted all data from the articles included, and the results were reviewed by the third reviewer. Inconsistencies between the reviewers were decided by a consensus. The published studies were examined in three steps: title, abstract, and full text.

### 4.5. Data Analysis

Prevalence of VRE was calculated and subgroup analyses were done according to the study region, isolate sources, and detection method. Considering the existence of heterogeneity in observational studies conducted in diverse settings, the random-effects model was used in determining the pooled prevalence of VRE which prompted the use of the DerSimonian and Laird method of meta-analysis [68,69].

### 4.6. Bias and Heterogeneity Analysis

The qualities of the study methods (study area, isolate source, and detection method) were used to assess the within-study biases. The across-study bias (small study effects) was examined by funnel plots. The heterogeneities of study-level estimates were assessed by Cochran’s *Q* test. Non-significant heterogeneity was accepted if the ratio of *Q* and the degrees of freedom (Q/df) was less than one. The percentage of the variation in prevalence estimates attributable to heterogeneity was measured by the inverse variance index (*I*^2^), and *I*^2^ values of 25%, 50%, and 75% were considered low, moderate, and high heterogeneity, respectively [69]. In this meta-analysis, the *I*^2^ value was high (96.26%) which is >75% an indication of significant heterogeneity. Due to this reason, the analysis was conducted using a random-effects model at 95% CI instead of the fixed-effects model. Funnel plot subgroup analyses were done if the heterogeneities were moderate to high. The sources of heterogeneity were analysed using the sensitivity analysis (leave-one-out meta-analysis), subgroup analysis, and meta-regression. Meta-analysis was performed using OpenMeta Analyst software version 10.10 [70].

## 5. Conclusions

We designed this study to obtain the pooled prevalence of VRE in Nigeria to provide baseline information to the National AMR Technical Working Group. The pooled prevalence of VRE in Nigeria was estimated at 25.3% (95% CI; 19.8–30.8%; *I*^2^ = 96.26%; *p* < 0.001) and *E. faecalis* is the most reported VRE. The prevalence of VRE is on the rise in Nigeria seeing the trend from the oldest to the earliest studies. High variability between studies may influence the estimate pooled prevalence at the national level. This can be overcome by using advanced diagnostic techniques in the detection of VRE and the implementation of a nationwide survey to estimate the true prevalence of VRE in Nigeria. This report indicates that a program directly targeting VRE nationally be in place and VRE be listed as a priority pathogen to reduce the burden of the infection. Adherence to proper and accurate antimicrobial usage, comprehensive testing and ongoing surveillance of VRE infections in the health care, community and environmental settings are required.

Availability of Data and Materials: The datasets used and/or analyzed during the current study are included in the manuscript.

## Figures and Tables

**Figure 1 antibiotics-09-00565-f001:**
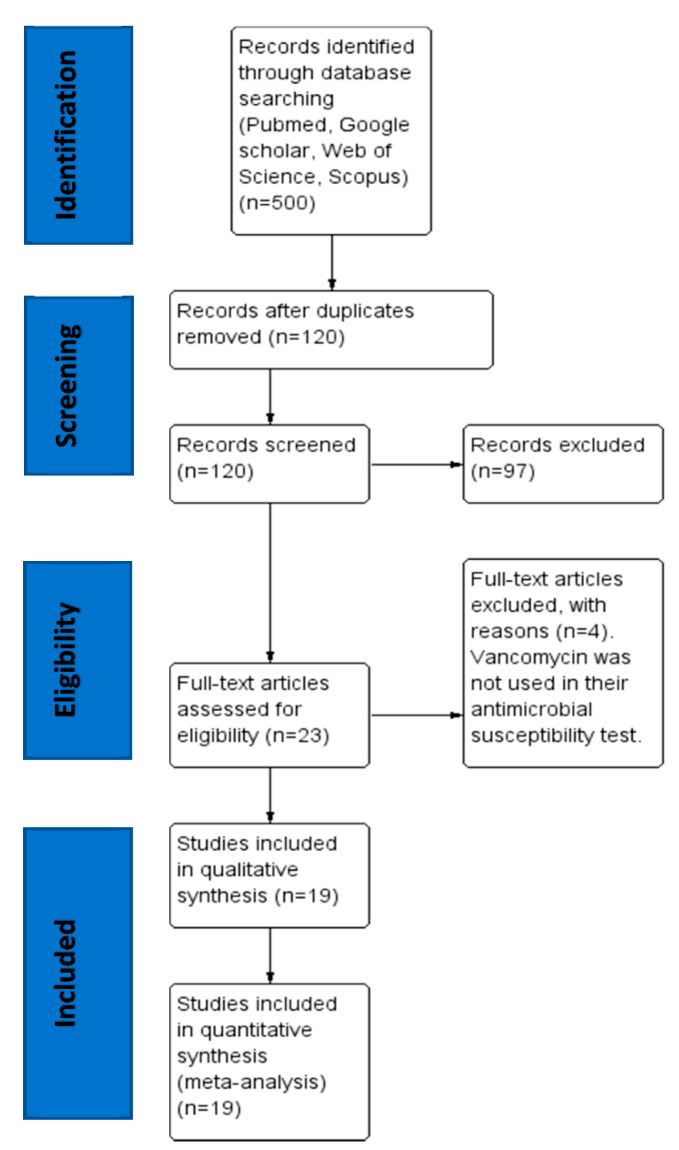
PRISMA flow diagram for the selection of eligible articles included in the study.

**Figure 2 antibiotics-09-00565-f002:**
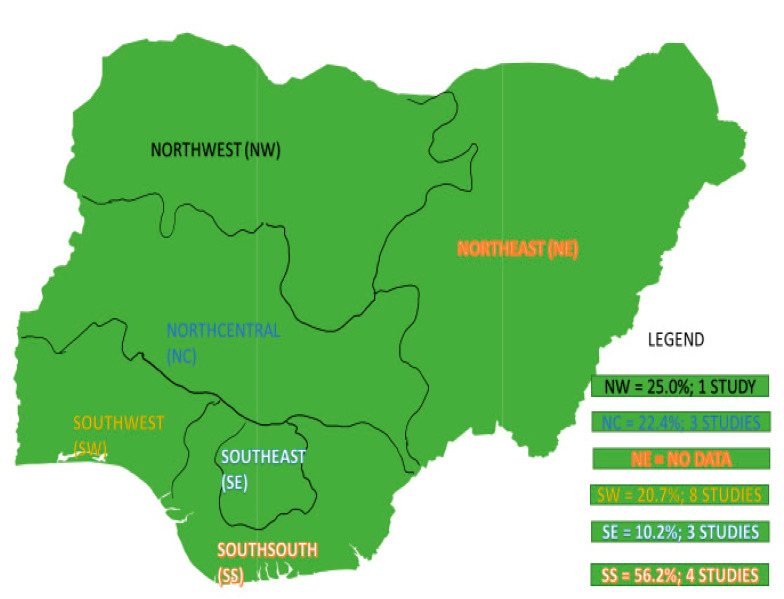
Spatial distribution and number of studies of VRE in Nigeria based on data extracted from eligible studies.

**Figure 3 antibiotics-09-00565-f003:**
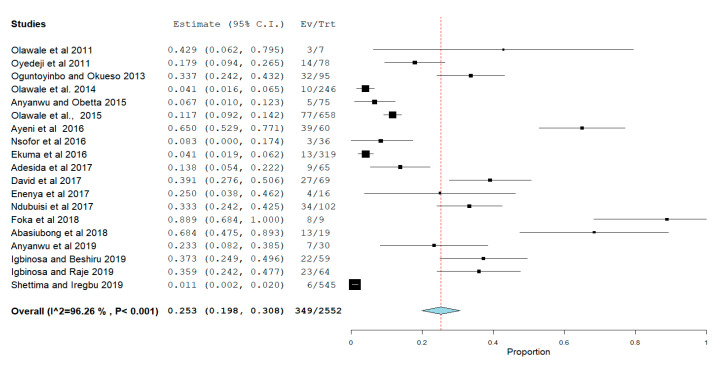
Forest plot showing the pooled prevalence of VRE in Nigeria.

**Figure 4 antibiotics-09-00565-f004:**
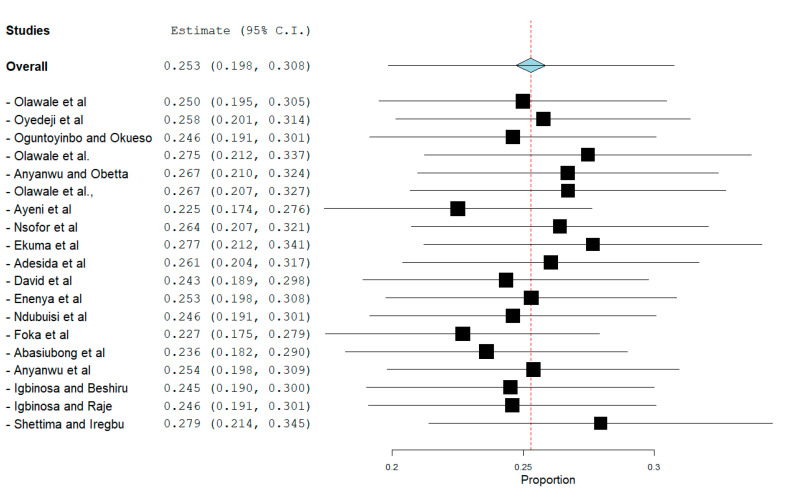
Leave-one-out forest plot of VRE in Nigeria.

**Figure 5 antibiotics-09-00565-f005:**
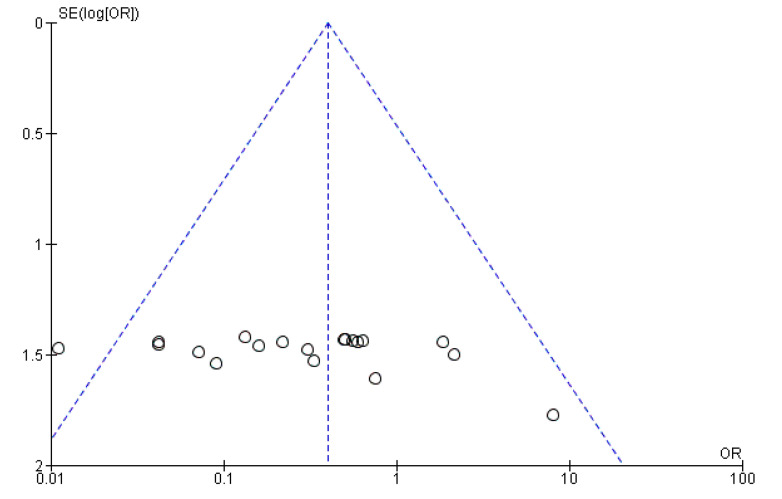
Funnel plot showing no publication bias.

**Table 1 antibiotics-09-00565-t001:** Characteristics of the selected studies reporting prevalence of Vancomycin-Resistant Enterococci (VRE) in Nigeria.

Author, Publication Year	Study Year	Study Area	Isolate Sources	Sample Size	Number Positive	Prevalence (%)	Detection Method
Olawale et al., 2011 [17]	2009	South-West	Clinical specimens	7	3	42.9	Disc diffusion
Oyedeji et al., 2011 [18]	2010	South-West	Environmental	78	14	18	Agar dilution
Oguntoyinbo & Okueso, 2013 [25]	2012	North-Central	Environmental	95	32	33.7	Disc diffusion
Olawale et al., 2014 [19]	2012	South-West	Environmental	246	10	4.1	Disc diffusion
Anyanwu & Obetta, 2015 [28]	2015	South-East	Animal	75	5	6.7	Disc diffusion
Olawale et al., 2015 [20]	2013	South-West	Environmental	658	77	11.7	Disc diffusion
Ayeni et al., 2016 [21]	2015	South-West	Animal	60	39	65	Disc diffusion
Nsofor et al., 2016 [29]	2016	South-East	Clinical	34	7	20.59	Disc diffusion
Ekuma et al., 2016 [22]	2013	South-West	Clinical	319	13	4.07	E test
Adesida et al., 2017 [23]	2017	South-West	Clinical	65	9	13.85	Disc diffusion
David et al., 2017 [24]	2017	South-West	Clinical	69	27	39.13	Disc diffusion
Enenya et al., 2017 [35]	2014	North-West	Environmental	16	4	25	Disc diffusion
Ndubuisi et al., 2017 [26]	2017	North-Central	Clinical	102	34	33.3	Disc diffusion
Foka et al., 2018 [31]	2018	South-South	Environmental	9	8	88.9	Disc diffusion
Abasiubong et al., 2019 [32]	2018	South-South	Clinical	19	13	68.4	Disc diffusion
Anyanwu et al., 2019 [30]	2018	South-East	Animal	30	7	23.3	Disc diffusion
Igbinosa & Beshiru., 2019 [33]	2018	South-South	Animal	59	22	37.3	Disc diffusion
Igbinosa & Raje, 2019 [34]	2017	South-South	Environmental	64	23	35.9	Disc diffusion
Shettima & Iregbu, 2019 [27]	2015	North-Central	Clinical	545	6	1.1	VRE Chromogenic agar

**Table 2 antibiotics-09-00565-t002:** Species distribution of VRE across studies.

Author, Publication Year	*E. faecium*	*E. faecalis*	*E. gallinarum*	*E. casseliflavus*	*E. mundti*	*E. hirae*	*E. dispar*	Total
Olawale et al., 2011 [17]	1	2	-	-	-	-	-	3
Olawale et al., 2014 [19]	-	10	-	-	-	-	-	10
Olawale et al., 2015 [20]	-	77	-	-	-	-	-	77
Nsofor et al., 2016 [29]	4	3	-	-	-	-	-	7
Ekuma et al., 2016 [22]	3	-	9	1	-	-	-	13
Adesida et al., 2017 [23]	6	3	-	-	-	-	-	9
David et al., 2017 [24]	-	27	-	-	-	-	-	27
Enenya et al., 2017 [35]	2	-	1	1	-	-	-	4
Ndubuisi et al., 2017 [26]	12	10	1	-	9	1	1	34
Igbinosa & Beshiru., 2019 [33]	13	8	1	-	-	-	-	22
Igbinosa & Raje, 2019 [34]	7	8	-	2	-	3	3	23
Shettima & Iregbu, 2019 [27]	3	-	2	1	-	-	-	6
	**51 (21.7%)**	**148 (62.98%)**	**14 (5.96%)**	**5 (2.13%)**	**9 (3.83%)**	**4 (1.70%)**	**4 (1.70%)**	**235**

**Table 3 antibiotics-09-00565-t003:** Subgroup analysis for comparisons of the prevalence of VRE across study regions.

Study Region	Number of Studies	Prevalence (%)	95% CI	*I*^2^ (%)	*Q*	Heterogeneity Test	
						DF	*p*
South-West	8	20.7	13.1–28.2	95.5	155.404	7	<0.001
North-Central	3	22.4	−3.6–48.3	97.81	91.227	2	<0.001
South-East	3	10.2	2.7–17.8	51.15	4.094	2	0.129
North-West	1	25.0	3.8–46.2	NA	-	-	-
South-South	4	56.2	33.5–79.0	88.37	25.802	3	<0.001
Overall	19	25.3	19.8–30.8	96.26	480.667	18	<0.001

**Table 4 antibiotics-09-00565-t004:** Subgroup analysis for comparison of prevalence of VRE among isolate sources.

Isolate Source	Number of Studies	Prevalence (%)	95% CI	*I*^2^ (%)	*Q*	Heterogeneity Test	
						DF	*p*
Clinical	8	19.9	12.5–27.2	195.18	145.362	7	<0.001
Environmental	7	27.2	17.3–37.2	95.33	128.519	6	<0.001
Animal	4	32.9	5.1–60.7	96.37	82.535	3	<0.001
Overall	19	25.3	19.8–30.8	96.26	480.667	18	<0.001

**Table 5 antibiotics-09-00565-t005:** Subgroup analysis for comparison of prevalence of VRE using various detection methods.

Detection Method	Number of Studies	Prevalence (%)	95% CI	*I*^2^ (%)	*Q*	Heterogeneity Test	
						DF	*p*
Disc diffusion	15	33.8	24.3–43.4	93.84	227.406	14	<0.001
Disc diffusion	1	4.1	1.6–6.5	-	-	-	-
Agar dilution	1	17.9	9.4–26.5	-	-	-	-
E test	1	1.9	1.9–6.2	-	-	-	-
VRE chromogenic agar	1	1.1	1.1–0.2	-	-	-	-
Overall	19	25.3	19.8–30.8	96.26	480.667	18	<0.001

**Table 6 antibiotics-09-00565-t006:** Subgroup analysis for comparison of prevalence of VRE across study periods.

Study Period	Number of Studies	Prevalence (%)	95% CI	*I*^2^ (%)	*Q*	Heterogeneity Test	
						DF	*p*
2009	1	42.9	6.2–79.5	-	-	-	-
2010	1	17.9	9.4–26.5	-	-	-	-
2012	2	18.5	−10.4–47.5	97.14	34.954	1	<0.001
2013	2	7.9	4.0–15.3	95.19	20.806	1	<0.001
2014	1	25.0	3.8–46.2	-	-	-	-
2015	3	23.1	0.7–45.5	98.19	110.342	2	<0.001
2016	1	8.3	−0.7–17.4	-	-	-	-
2017	4	30.2	18.0–42.3	82.92	17.560	3	<0.001
2018	4	53.6	26.5–80.7	90.54	31.711	3	<0.001
Overall	19	25.3	19.8–30.8	96.26	480.667	18	<0.001

**Table 7 antibiotics-09-00565-t007:** Final multivariable meta-regression model.

Variable	Coefficient	*p*-Value	95% CI
**Study area**			
South-West	Reference		
North-Central	0.175	0.005	5.2–29.8
North-West	−1.044	<0.001	−142.1–−66.7
South-East	−0.533	<0.001	−65.9–−40.7
South-South	−0.286	0.003	−47.7–−9.4
**Isolates source**			
Clinical	Reference		
Animal	0.273	0.188	−13.3–67.9
Environmental	0.865	<0.001	38.4–134.6
**Detection method**			
Disc diffusion	Reference		
Agar dilution	−1.114	<0.001	−143.7–−79.2
E test	0.371	<0.001	16.7–57.5
VRE chromogenic agar	−0.917	<0.001	−113.9–−69.6
**Study period**			
2009	Reference		
2010	−1.132	<0.001	−147.8–−78.6
2012	−1.177	<0.001	−148.9–−86.5
2013	−0.093	0.219	−24.1–55.0
2014	−0.230	0.041	−45.1–−0.9
2015	−0.688	<0.001	−87.9–−49.6
**Constant**	0.429	0.022	6.2–79.5

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
