# Peer review of "Vancomycin-Resistant Enterococci (VRE) in Nigeria: The First Systematic Review and Meta-Analysis"

_antibiotics, 2020, doi:10.3390/antibiotics9090565_

Round 1
Reviewer 1 Report
This is a valuable contribution on Vancomycin-Resistant Enterococci (VRE) treatment patterns In Nigeria.
The Systematic Review and Meta-Analysis were conducted via PRISMA guidelines.
It fills an important knowledge gap and thus is worthy of publishing.
Yet its main weakness is insufficiently heterogeneous evidence base.
It should be significiantly expanded.
Thus I warmly recommend consideration for inclusion of at least several of the beneath mentioend sources:
Jakovljevic, M., Jurisevic, M., & Mouselli, S. (2018). Antibiotic resistance in Syria: a local problem turns into a global threat. Frontiers in Public Health, 6, 212.Front. Public Health, 02 August 2018 | https://doi.org/10.3389/fpubh.2018.00212 Zhussupova, G.; Skvirskaya, G.; Reshetnikov, V.; et al. The Evaluation of Antibiotic Consumption at the Inpatient Level in Kazakhstan from 2011 to 2018. Antibiotics 2020, 9, 57. Troeger, C., Blacker, B., Khalil, I. A., Rao, P. C., Cao, J., Zimsen, S. R., ... & Adetifa, I. M. O. (2018). Estimates of the global, regional, and national morbidity, mortality, and aetiologies of lower respiratory infections in 195 countries, 1990–2016: a systematic analysis for the Global Burden of Disease Study 2016. The Lancet infectious diseases, 18(11), 1191-1210. Ehinmidu, J. O. (2003). Antibiotics susceptibility patterns of urine bacterial isolates in Zaria, Nigeria. Tropical Journal of Pharmaceutical Research, 2(2), 223-228. Robertson, J., Iwamoto, K., Hoxha, I., Ghazaryan, L., Abilova, V., Cvijanovic, A., ... & Dzhakubekova, A. (2019). Antimicrobial medicines consumption in Eastern Europeand Central Asia–an updated cross-national study and assessment of quantitativemetrics for policy action. Frontiers in pharmacology, 9, 1156. Jakovljevic, M., Jakab, M., Gerdtham, U., McDaid, D., Ogura, S., Varavikova, E., ... & Getzen, T. E. (2019). Comparative financing analysis and political economy of noncommunicable diseases. Journal of medical economics, 22(8), 722-727. Sapkota, A. R., Coker, M. E., Goldstein, R. E. R., Atkinson, N. L., Sweet, S. J., Sopeju, P. O., ... & Shireman, L. (2010). Self-medication with antibiotics for the treatment of menstrual symptoms in southwest Nigeria: a cross-sectional study. BMC public health, 10(1), 610. Jakovljevic, M., Timofeyev, Y., Ranabhat, C. et al. Real GDP growth rates and healthcare spending – comparison between the G7 and the EM7 countries. Global Health 16, 64 (2020). https://doi.org/10.1186/s12992-020-00590-3 Okonko, I. O., Donbraye-Emmanuel, O. B., Ijandipe, L. A., Ogun, A. A., Adedeji, A. O., & Udeze, A. O. (2009). Antibiotics sensitivity and resistance patterns of uropathogens to nitrofurantoin and nalidixic acid in pregnant women with urinary tract infections in Ibadan, Nigeria. Middle-East J. Sci. Res, 4(2), 105-109. Assuming authors willingness to adopt at least several of these recommendations alongside some at their own disposal I would be willing to rview revised manuscript assuming its maturity for publishing.Author Response
Heterogenous evidence base should be significantly expanded
Response: Variables we added as our possible source of heterogeneity are those we could extract from each study and has met the inclusion criteria. We would take note of this point for future studies.
Inclusion of several sources mentioned:
Response: Five of the recommended sources was added. Reference number [57, 62-65].
Reviewer 2 Report
I have read with interest this systematic review and meta-analysis about the pooled incidence of VRE in clinical, animal, and environmental samples of enterococci in Nigeria. The manuscript is well written. I have few comments and suggestions for improvement.
- The authors should report the incidence of VRE by species (E. faecalis vs E. faecium) from the available literature. This should be attempted even if not all included studies reported this information.
- Abstract, first sentence: Enterococci “are” not “is”.
- Introduction: The authors mention that enterococci are most common causes of UTI, inflammation …. It should be “common” not “most common”.
- Introduction: MDR implies resistance to 3 classes of antibiotics. VRE are not considered MDR since they are only required to be resistant to one class.
- Introduction, lines 54-57: Duplicate sentence is incorrect. Please delete.
- Introduction: Authors mention that faecalis exhibit high resistance to vancomycin. This is not accurate. Only E. faecium does that.
- Introduction, line 77: The sentence about coronavirus is out of context. Delete.
- Line 117: References should be listed at the end of the sentence.
- Table 1: The last 3 studies in the table are not formatted like the prior ones. Revise.
- Figures 4, 6, 7, 8, and 9 can be safely deleted without compromising content.
- Table 2, second line and Table 5, third line: 95% CI for incidence should not be under zero.
- Table 4 may be deleted and explained in text.
- Conclusion: The opening sentence about COVID-19 is out of context. The conclusions should be derived from current results.
- The pooled incidence of VRE should be listed in the conclusions.
Author Response
Report incidence of VRE by species
Response: We report the incidence of VRE by species in Line 119. We also have added the result to the abstract and conclusion section.
Enterococci “are” not “is”:
Response: Corrected. Line 13
“common” not “most common”.
Response: Corrected. Line 39
VRE is not considered MDR since they are only required to be resistant to one class.
Response: MDR has been deleted and sentence rephrased. Line 44
lines 54-57: Duplicate sentence is incorrect
Response: Duplicate has been deleted and sentence rephrased. Line 54
faecalis exhibit high resistance to vancomycin. This is not accurate. Only E. faecium does that.
Response: What we meant was that in comparison to E. gallinarum and E. casseliflavus, they both have a higher resistance with E. faecium being the most resistant.
The sentence about coronavirus is out of context. Delete.
Response: Deleted.
References should be listed at the end of the sentence.
Response: Listed. Line 117
Table 1: The last 3 studies in the table are not formatted like the prior ones. Revise.
Response: Revised. Line 117, Table 1
Figures 4, 6, 7, 8, and 9 can be safely deleted without compromising content.
Response: Figures 6,7,8 and 9 are deleted. We did not delete Figure 4 because we felt it shows an important component of the meta-analysis.
Table 2, second line and Table 5, third line: 95% CI for incidence should not be under zero.
Response: It was a source of worry for us as well, but because we carefully repeated the analysis and got the same result, we decided to go with as such. We thought perhaps, it was because of the lack of enough data.
Table 4 may be deleted and explained in the text.
Response: Since we will be deleting Figure 8 which shows the forest plot of subgroup analysis by detection method, we felt we don't have to delete Table 4 (Now Table 5) since it shows the Subgroup analysis for comparison of prevalence of VRE using various detection methods.
Conclusion.
Response: Opening context about Covid-19 has been deleted, Line 397. The pooled incidence of VRE has been listed in the conclusions, Line 399.
We thank you for your thorough and insightful review. You have indeed improved the quality of this work. Thank you and best regards.
Reviewer 3 Report
The material and methods section must be inserted after the introduction and before the results.
Please enter the time range covered by the found studies to produce the meta-analysis both in the abstract and in the materials and methods section.
In the results, in particular in the summary tables (such as Table 1,2,3,4,5) the period in which the study was conducted is not indicated (months). you could insert it together with the others results.
Author Response
Materials and methods section
Response: We followed the Journal template where materials and methods come after the result and discussion section.
The time range in the abstract and materials and methods section
Response: We did not include the time range in the materials and methods section because we wanted to get as many studies as possible and it was not part of our inclusion or exclusion criteria. We, however, included the time range in the Result section (Table 1; Study period). The time range (Study period) has been added in the abstract section (Line 22).
The month in which the studies were conducted in tables 1,3,4,5,6, was not indicated
Response: We did not include months in Table 1 because most of the studies did not report it. Table 3,4,5 and 6 did not show period (month) because it is a subgroup meta-analysis for all variables listed in the study.
We thank you for your insightful comments and review. Best regards.
Round 2
Reviewer 2 Report
I thank the authors for appropriately revising the manuscript. I have no further comments.